# Preliminary Clinical and Radiographic Evaluation of a Novel Resorbable Implant of Polylactic Acid (PLA) for Tibial Tuberosity Advancement (TTA) by Modified Maquet Technique (MMT)

**DOI:** 10.3390/ani11051271

**Published:** 2021-04-28

**Authors:** Victoria Valiño-Cultelli, Óscar Varela-López, Antonio González-Cantalapiedra

**Affiliations:** Department of Anatomy, Animal Production and Veterinary Clinical Sciences, Veterinary Faculty, Universidad de Santiago de Compostela, 27002 Lugo, Spain; oscar.varela@usc.es (Ó.V.-L.); antonio.cantalapiedra@usc.es (A.G.-C.)

**Keywords:** TTA, tibial tuberosity advancement, MMT, modified maquet technique, polylactic acid, PLA, 3D print, scaffold, dog

## Abstract

**Simple Summary:**

The metallic implants carry many problems as infections, bone resorption, pain at the site, etc. Removing these metallic implants is an expensive procedure, and it entails risks. Due to the costs, the most common in veterinary medicine is to remove metallic implants only when a problem occurs. The rupture of the cranial cruciate ligament (RCrCL) is the most common orthopedic pathology in dogs and the most frequent cause of arthrosis, pain, and limp on the knee joint. Those are the reasons that encouraged us to develop a new type of biodegradable implant comprised of polylactic acid (PLA). This is a non-toxic material that can be eliminated by natural metabolic pathways and use the PLA implant in tibial tuberosity advancement (TTA), which is a technique for the resolution of RCrCL in dogs. In our study, PLA implants for TTA provide good functional results, presenting an acceptable number of complications. The implants show a faster ossification than metallic implants, which was not affected by age or by body weight. The PLA implants have a clinical recovery time similar to metallic implants.

**Abstract:**

Our objectives were to determine whether PLA implants can be used in TTA with successful results; secondly, to observe whether they provide a faster bone healing; finally, to determine whether weight or age influences bone healing scores. PLA cages were created with a 3D printer. TTA by MMT with PLA implants was performed in 24 patients. Follow-ups were carried out pre-surgical, at 1, 2, and 5 months and consisted of a radiographic study and a lameness assessment. A comparison was performed in terms of weight and age. Patients data, time between follow-up examinations, healing score, and lameness score were compared between patients using commercial software for statistically significant differences *p* < 0.05. Eighteen dogs finished the study. The ossification degrees presented statistically significant differences between each other. PLA implants maintained the advancement in 100% of cases. Comparing weight and age did not present any statistically significant differences between groups. Lameness presented statistically significant differences between follow-up examinations. Complications were observed in 20.8%. PLA implants for TTA provide good functional results, presenting an acceptable rate of complications. They provide a faster bone healing of the osteotomy gap, which was not affected by age or body weight, and have a clinical recovery time similar to metallic implants.

## 1. Introduction

The rupture of the cranial cruciate ligament (RCrCL) is the most common orthopedic pathology in dogs [1,2,3] and the most frequent cause of secondary degenerative arthrosis, pain, and limp on the knee joint [4,5]. Damage in the cranial cruciate ligament (CrCL) produces synovial membrane inflammation, articular cartilage injury, production of osteophytes, subchondral bone degeneration, and periarticular soft tissue fibrosis [3]. The treatment for CrCL is surgical, and its objective is to stabilize the stifle joint by neutralizing the tibiofemoral shear forces [6].

The surgery techniques that obtain better results are the ones that modify the proximal tibial geometry [7,8]. The CrCL is not replaced, instead of that the neutralization of tibio-femoral forces allows for the dynamic stability of the knee and does not allow its subluxation with weight-bearing [9]. Among this type of technique, there is the tibial tuberosity advancement (TTA) [10], which neutralizes cranial tibiofemoral shear force by advancing the insertion of the patellar ligament until it is perpendicular to the tibial plateau with the joint in extension. This 90º patellar tendon angle (PTA) is typically maintained by stabilizing the osteotomy of the proximal tibia with procedure-specific stainless steel or titanium plate and cage implants [10]. These implants avoid the osteotomy collapse, allowing for the maintenance of the desired PTA, although once the osteotomy is healed, they are not needed anymore.

Furthermore, the bone, in response to the surrounding mechanical stimuli, adapts its anatomical structure through natural growth and resorption processes [11]. In addition to that, metallic implants cause higher stiffness than bone tissue, as well as mechanical stress shielding in bone inducing its resorption, ultimately leading to osteolysis or prosthesis loosening [12]. However, there are some types of metallic implants, such as titanium, that integrate very well to the bone and induced their osseointegration [13,14,15].

In human medicine, implant removal is a common practice [10], avoiding the osteolysis and other problems that this may cause [16,17], although removing them is an expensive [18] procedure and entails risks [19]. In veterinary medicine, metallic implants are removed only when a problem occurs [17].

The TTA plate and cage function together as a tension band wire construct, the cage preventing the collapse of the osteotomy site and loss of desired patellar tendon angle (PTA) while the plate neutralizes distractive forces [20]. Once the osteotomy site is healed, the implants are not necessary anymore, so they have only a temporary role in maintaining TTA stability and PTA [20]. However, removing TTA implants is not an easy technique; it can cause the fracture of the tibial diaphysis or tuberosity and is expensive for the owners [20,21].

Moreover, there are many studies on TTA that describe many types of techniques [22]; we decided to use the modified maquet technique (MMT) for TTA, doing the fixation with a pin and a tension band wiring.

Taking this into account, the development of resorbable implants has been increasing over the last years. One of the most used materials in tissue engineering is PLA, which is an aliphatic polyester derived from lactic acid that exists in nature; it is biodegradable, thermoplastic, with great mechanical strength, and has excellent biocompatibility [9,23]. Degradation products from PLA are not toxic and are eliminated by natural metabolic pathways [24]. In addition, 3D printing implants have been converted into one of the best surgical options, allowing to obtain personalized implants [23].

Bearing all this in mind, one of our purposes is to determine whether PLA resorbable implants (Figure 1), custom-made in a 3D printer, can be used in TTA by MMT with successful results, being able to develop them as an alternative to metallic implants. The second objective is to observe whether they provide a faster bone healing, and consequently, faster recovery of the patient. Finally, we also want to determine whether weight or age affects bone healing.

## 2. Materials and Methods

### 2.1. Design and Fabrication of Implants

For the fabrication of implants, the fused deposition modeling (FDM) technique of 3D printing was used, employing a 3D printer Anet 8, clone’s Prusa i3 (Dot Go 3D Technology Corporation, Xiangtan, China). The design was carried out using Sketchup software (Trimble Inc., Limited Sunnyvale, California, EEUU) as a scaffold with a polygonal wedge form with an internal grating form (Figure 2). Its digital dataset was saved as a stereolithography file (STL). The width, depth, and length of the implant were calculated according to the proximal tibia anatomy (Figure 3). Slice software Cura 2.3 was used to generate G code for the printer, and the printing software was Repetier Host V 2.0.1 (Hot-WorldGmBH and Co., Knickelsdorf, Germany).

The homogeneous distribution of the filler is crucial for the mechanical properties because it can favor uniform stress distribution over the whole implant [25]. In addition, the inner grid design of the scaffold allows introducing different materials and has osteoconductive properties.

The scaffolds were comprised of melt medical-grade PLA (Leon 3d, Valverde de la Virgen, Leon, Spain), extruded through a heated metal nozzle (0.4 mm diameter) at 210 °C with a layer height of 100 µm. Once the implants were ready, the sterilization was performed in an autoclave with a conventional rubber program.

A compressive test was performed on the implants in a static mechanical testing device, instron microtest press “EM1/10/FR/SCM”, according to ISO 604:200 [26], with a force transducer “TSC-1/10K” head for 1000 N, with a device designed for the adaptation of the plates to the polygonal wedge form, and a compression speed of 20.000 mm/min.

### 2.2. Clinical Trial

#### 2.2.1. Selection of Patients

This prospective randomized study was conducted from December 2017 to December 2019 in the Rof Codina University Veterinary Hospital (Lugo, Spain).

The surgery with new implants was performed in 24 owned dogs. All owners were informed about the new procedure and signed a consent form allowing all documentation regarding their dog to be used for scientific research and publication.

Dogs were included in the study if diagnosed with RCrCL, based on a history of hind limb lameness with stifle pain, articular effusion, and inflammation, and confirmed by positive cranial drawer motion and/or positive cranial tibial thrust [27,28] upon orthopedic examination and supportive radiographic evidence of stifle effusion or osteoarthritis. All dogs in this study were skeletally mature. In each case, the surgery was performed by the same expert surgeon and following a standardized protocol.

Patients were excluded if they had another illness different from RCrCL, had been subjected to any other surgical intervention on the affected knee and/or a concurrence pathology in the stifle, different from RCrCL. The patients that did not complete the protocol were also excluded, although their follow-ups were conducted by phone call.

#### 2.2.2. Protocol

An anamnesis, physical and traumatological exploration, pre-surgical radiographs, complete blood count, and biochemistry profiles were determined for each patient. Radiographs were used for the determination of cage size by the PTA-TP method [29].

On the day of the surgery, the premedication for anesthesia was 10 µg/kg intramuscular (IM) medetomidine and 0.3 mg/kg IM morphine. For the induction, we used 2 mg/kg intravenous (IV) propofol, and during the procedure, sevoflurane was employed. Analgesia was obtained by 0.2 mg/kg IV meloxicam and a continuous infusion rate of morphine-lidocaine-ketamine IV. We used cefazolin 22 mg/kg IV as an antibiotic, one dose 30 min prior to the surgery and another dose 60 min after the first incision.

Once the patient was under premedication effects, the hindlimb was prepared aseptically, and when the patient was under anesthesia effects, the standard medial approach for the tibia was used [6]. Next, the bicortical osteotomy was performed perpendicular to the sagittal plane of the tibia, starting at the proximal cortical surface and leaving 1 cm from the distal cortical surface. Then the cranial distraction of the tibia was performed using a spacer attached to a T-handle in a careful manner in order to avoid causing a fracture of the distal cortical surface. The PLA cage was placed into the osteotomy site at the proximal extent of the osteotomy, 1–3 mm distal to the tibial plateau (Figure 4). Once the implant was in a correct position, it was fixed with a 1.5 mm pin, the pin was placed under de tibial insertion of the patellar ligament with a drill, from ventral to dorsal direction (to avoid the pin’s loosening with the patient movements in a future), the pin went through the tibial crest, the implant, and the tibial diaphysis. After the pin was placed, we folded and gently tapped it to impact it against the tibial crest (to avoid the pin’s rotation in the future); we also used a tension band wiring between the diaphysis and the tibial crest for more fixation and neutralization of distractive forces. For the tension band wiring positioning was performed a hole in the distal portion of the osteotomized tibial crest, leaving at least 1 cm proximal to the end of the osteotomy site, and another hole in the tibial diaphysis, a bit lower than the first one; into the holes, it was performed with a figure-of-eight wire with a knot on each side (to distribute tension forces equally to both sides). The procedure and its result are illustrated in Figure 4.

The limb was evaluated in order to confirm the absence of positive cranial drawer motion and/or positive cranial tibial thrust. Finally, the closure of the surgical site was carried out as is described by Lafaver (2007).

Before the patient was awake, we obtained postsurgical radiographs to assess whether the implants were correctly placed.

Once the patient was awake, the analgesic continuous infusion rate was continued until discharge, the same day in the afternoon. Home treatment was the same for all patients, that is, meloxicam 0.1 mg/kg PO q 24 h for 9 days, cefazoline 22 mg/kg PO q 8 h for 10 days, digestive protection (depending on patient’s weight) for 10 days. In addition, a Robert-Jones bandage was applied from the day of the surgery up to 4 days after, and limited exercise was recommended until follow-up examination.

In case the patient presented symptomatic lameness in one of the follow-up examinations, the treatment was meloxicam 0.1 mg/kg PO q 24 h for 7 days.

#### 2.2.3. Data Collection

The data collected were sex, age, body weight, breed, lifestyle, level of exercise difficulty presented by the patient, dates of follow-up radiographs, TTA cage size, and complications.

If there were complications, they were assessed as a major or minor complication, according to previous publications [30]. Those owing to the pin and tension band wiring removal, which was planned in case the patient needed them, were not considered as such.

The follow-up was carried out by complete physical examination and radiographs at 1, 2, and 5 months after surgery. All the radiographs were obtained in laterolateral and caudocranial views and using the same X-ray equipment. In addition, all the observations described by the owner were also recorded.

#### 2.2.4. Radiographic Assessment

Only lateral radiographic projections at 135º were taken into account in order to assess the osteotomy site healing. All patients were randomly assigned a number, and their radiographs had the same number; this evaluation was performed independently by two observers using the commercial software OsiriX MD 11.0. (PIXMEO SARL, Geneva, Switzerland) (open-source software; accessed date: 23 January 2021 www.osirixviewer.com).

The evaluation of the osteotomy site healing was conducted using a score on a scale from 0 to 4, according to a previously published study [31], which was adapted for our technique. The sites were defined as the region of osteotomy proximal to the cage, the region of the cage, and the region of osteotomy distal to the cage. A 0–4 scale was used, with 0 indicating no osseous healing; 1 representing early bone production without bridging between the tibial tuberosity and the shaft of the tibia; 2 indicating bridging bone formation at one site; 3 indicating bridging bone at two sites; 4 representing bridging bone at all three sites. Moreover, for the correlation between bone healing and weight, patients were divided into two groups, more than 30 kg (8 dogs) and less than 30 kg (10 dogs) for the different follow-up examinations. The same was performed in terms of age, being separated into two groups, more than 7 years old (dogs considered senior [32]) and less than 7 years old.

The maintenance of advancement was assessed in each radiograph.

Follow-up complications and observations were also described.

#### 2.2.5. Lameness Assessment

A numerical rating scale, previously published by Etchepareborde (2011), was used for assessing the lameness in each of the patients [33]. This scale had six levels of lameness severity: 0 = no detectable lameness at a walk or trot and no detectable lateral weight shift at a stance; 1 = no detectable lameness at a walk or trot, and minor lateral weight shift at a stance; 2 = lameness at a walk or trot without hip hike; 3 = lameness at a walk or trot with hip hike; 4 = non-weight-bearing at a trot; 5 = non-weight-bearing at a stance [33]. The degree of the lameness was evaluated independently by three observers in the first visit of the patient (previous to the surgery) and after the surgery when they came for radiology tests.

### 2.3. Statistical Method

The ossification degrees were statically compared in the different follow-up examinations. The patients were divided into 3 groups, group 1 (1st month follow-up), group 2 (2nd month follow-up), and group 3 (5th month follow-up). Results were expressed as mean ± standard deviation, and the statistical analysis was carried out with Sigma Plot 12.5 (Systat software Inc., San Jose, CA, USA). The variables were not normal; thus, the statistical comparison was performed with the Kruskal–Wallis test, whereas for the post-hoc analysis Tukey test was used.

In terms of age influence, in the different follow-up examinations, the patients were divided into two groups, group 1 (<7 years old) and group 2 (<7 years old). We compared these groups with the ossification groups with the T-test. The same procedure using the same test was carried out, taking into account the influence of different body weights, group 1 (<30 kg) and group 2 (>30 kg).

Patients’ levels of lameness before the surgery and for each of the follow-ups were compared using the Kruskal–Wallis test.

For all the variables, the differences were considered statistically significant when *p* < 0.05.

## 3. Results

This study included 24 skeletally mature patients that received a new PLA implant, 18 of them completed the study, whereas six of them did not.

Among the 18 dogs that completed it, there were five crossbreeds, four golden retrievers, two boxers, one French bulldog, one beagle, one Spanish mastiff, one German shepard, one cocker, one epagneul breton, and one galician laxeiro. There were eight males, out of whom two were neutered and 10 females, out of whom two were neutered. The mean age at the moment of the surgery was 72 ± 34.7 months (ranging between 21 and 132 months) with a medium bodyweight of 26.67 ± 12 kg (ranging between 8.4 and 55 kg). The median cage size used for the surgery was 8.94 ± 1.92 mm (ranging between 5 and 13 mm). A total of 55.55% of the surgeries were performed on the left knees and 44.45% on the right knees.

The results of the compression test had a maximum load of 10,100 ± 50 N and a compressive strength of 1,286,200 N/m^2^ ± 6180 N/m^2^.

The mean time of the radiologic follow-ups was 35.22 ± 10.72 days for the first one, 71.88 ± 9.39 days for the second, and 154.27 ± 22.02 days for the last one (Table 1) (Figure 5).

The mean values for ossification degrees at follow-ups were 1.33 ± 1.13, 2.61 ± 0.84, and 3.55 ± 0.51, respectively. The ossification degrees presented statistically significant differences between each other (*p* < 0.05) when comparing the results between 1st and 2nd follow-ups, 1st and 3rd follow-ups, and 2nd and 3rd follow-ups (Table 2).

When comparing the body weight between the two groups (more than 30 kg (8) vs. less than 30 kg (10)), the ossification degrees still did not present any statistically significant differences at follow-ups. The same occurred with the influence of age on the ossification degrees, no statistically significant differences being found between dogs younger than 7 years old (12) and older than 7 (6) at any of the follow-ups (Table 3).

In addition, we also measured all the radiographs to make sure that the advancement of the tibial crest was maintained over time; in 100% of the dogs, the advancement was maintained at 5 months following the surgery. No patient presented complete reabsorption of the PLA implant.

Regarding the lameness degrees at the pre-surgical assessment (ps), the mean value was 3.55 ± 0.92; at first follow-up (1st) it was 2.05 ± 1.39; at the second follow-up (2nd) it was 0.83 ± 0.98; and at the final follow-up (3rd) it was 0.11 ± 0.32. The lameness degrees presented statistically significant differences between follow-up examinations: ps vs. 2nd; ps vs. 3rd; 1st vs. 3rd (*p* < 0.05) (Table 4).

Complications were observed in 20.8% (5) of the dogs that had continued the study for 5 months after surgery, including those with follow-up by phone, two minor complications, and three major complications [30] (Table 5). In two dogs (8.3%), the pin was removed, although this was not considered as a complication because the owners were notified about that possibility before surgery.

## 4. Discussion

The present study was performed with 24 owned dogs, out of whom 18 completed it, and the others were monitored by phone call. The results of the technique implemented, the ossification degrees, and the solution of the lameness were acceptable in all of them, which supports our first hypothesis.

The population in the study was effectively randomized (not considering sex, age, body weight, breed, lifestyle, level of exercise difficulty presented by the patient, TTA cage size, and complications for the inclusion in this study), which made it possible to compare the ossification in patients of different weights and ages, which is something that was not possible in other similar studies [34], thus demonstrating that these parameters do not affect the ossification degree, which was our last hypothesis. In addition, considering that this was a prospective study, the radiographic follow-ups were planned, so there were small differences in terms of follow-ups days between the patients, which is something that may affect the evaluation of the ossification healing [35].

The decision to use MMT was reached because it has been proven that the fixation only with a pin and a wire tension band eliminated stress risers created by the plate, fork, and screws because the osteotomized piece of bone was stressed by the drilled holes for the placing of the plate [36]. Some in vitro studies support the fact that this technique increases the strength of the fixation and decreases the complication rate of TTA [22]. In addition, our surgeon mentioned that this technique was easier to perform and faster than the conventional TTA technique, so it may have a lower learning curve, but further studies are needed in order to confirm it.

Moreover, a study with 60 kg goats showed that the patellar force during standing was 207 N and 1000 N during a trot, whereas the force to failure of the goat patellar tendon was 3100 N [37]; in another in vitro study in dogs, for the previous described MMT with metallic implants, the force to failure of the dog patellar tendon was 1.47 N [22]. In our study, the implant compression test had a maximum load of 10,100 ± 50 N and a compressive strength of 1,286,200 ± 6180 N/m^2^, so one would expect that in vivo circumstances, the patellar tendon may fail before the implant could collapse. Furthermore, in our study, there’s no evidence of implant collapse, but a study with a larger number of patients may be necessary to confirm it.

Regarding the implant osteoconductivity and osteointegration, contrasting the results obtained with other similar studies where TTA was applied using bone grafts, the ossification degree in our study is lower than that obtained in a previous paper [34] and greater than the results obtained in other studies [31] where the ossification with metallic implants was analyzed, although these studies used bone grafts. Therefore, in contrast to another previous publication by Hoffmann where the ossification degrees for titanium TTA were analyzed, 87% of the dogs in the study reached a certain degree of ossification (grade 2 or higher) at an average of 8.6 weeks postoperatively [31], in our study 88.8% of the patients reached a certain degree of ossification (grade 2 or higher) at 8 weeks.

In addition, this study was contrasted to other similar studies that compared biodegradable implants with metallic implants without using bone grafts [17,20], our results being better. This supports part of our second hypothesis that our resorbable implants provided faster bone healing.

Some studies mentioned that biomaterials may provide faster recovery and ossification in bone defects because the implant reabsorption induces the progressive loss of the osteotomy support and increases the tension, which stimulates the mineralization and bone production [17,38]. In our study, we obtained the healing of the bone at two sites, which stabilized the implant and prevented its failure at the 2nd follow-up for 61.1% of the patients.

There are many factors that may affect the ossification degrees. The most important ones are the patient’s age (the older the patient, the ossification degree decreases) and the time between radiographic follow-ups [17]. Other factors that may affect are weight: the heavier the patient the ossification degree decreases and the risk of complications increases [31,39,40,41]. The results in our study suggest that with this type of implant, there are no differences between different patients, supporting our third hypothesis, although further studies with a higher number of patients are recommended.

Complete implant reabsorption was not observed in any case, neither was it recorded in other studies on degradable implants that were conducted over a shorter period of time and used similar materials [17,20,42]. There are a few studies about that in laboratory animals, and in one of them, the period of degradation was between 6 and 12 months [43]; in another study conducted on rodents, it was mentioned that the degradation was completed in 24 months without inflammatory reaction [44].

Some studies observed that the degradation of PLA produced lactic acid that may cause acidification of the extracellular environment with a consequent inflammation affecting cellular behavior [45]. However, another study noted that there were no cytotoxic effects caused by PLA on human cells [23]. In addition, many authors agree about the release of ions to the environment by PLA, contributing to the proliferation and differentiation of cells into osteogenic phenotypes [46,47,48,49,50]; it was therefore hypothesized that the release of such ions might contribute to triggering and enhancing bone formation [24]. Moreover, it was mentioned that the augmentation of acid products may lead to the formation of osteolytic zones in cancellous bone [51,52,53].

Despite the fact that our study does not have any histological evaluation yet, we did not observe any significant macroscopic inflammatory radiographic changes or signs of inflammation by exploration. No osteolytic zones were observed in bone radiographs. Our aim is to continue to take follow-up radiographs in order to determine when implant reabsorption occurs and the presence of cellular inflammatory reaction, if any.

PLA is a resorbable material, and a point that concerned certain authors was whether it could maintain the osteotomy advance, which is the theoretical basis of the TTA technique [20]. As explained in a previous report on resorbable materials, the loss of the osteotomy support is progressive [17]; the advancement in our study was maintained in all the patients over time at follow-ups, which means that the new implant has an acceptable function maintaining TTA’s advancement, thus supporting our first hypothesis.

Regarding lameness, its values improved in all patients, the results of the technique were functionally acceptable, having a final mean of 0.11 ± 0.32 (keeping in mind that 0 means no detectable lameness at a walk or trot and no detectable lateral weight shift at a stance). In addition, results suggested that recovery to normal function is gradual. A total of 100% of the dogs had no detectable lameness at the last follow-up, 88.89% with normal weight-bearing at stance. These results are consistent with those of other similar studies that observed no presence of lameness at 12 weeks for 95.38–98.46% of the patients [54,55]; the results are better than those obtained in a previous paper that mentioned 68% resolution of lameness at 12 weeks [56]; all these studies were conducted with metallic implants. Regarding weight-bearing, another study observed that at 12 weeks, 80% of the patients had normal weight-bearing standing [55]; other two previous studies noted that TTA improved weight-bearing but not always restored it completely [57,58]. Muscle atrophy occurred following RCrCL and may progress after surgical intervention [59,60,61], the reason being the limitation in exercise performance that is needed for the stabilization and integration of the implant before the patient can return to normal activity [62,63]. It has been proven that in dogs with iatrogenic RCrCL and immediate stifle stabilization, muscle atrophy was evident by 2 weeks and progressed until 5 weeks postoperation and a slight recovery in muscle mass was evident at 10 weeks [61], besides that small changes in muscle mass correlated with significant changes in muscle strength [59]. That explains the reason why we did not see any significant change in lameness between de pre-surgical and the first follow-up (4 weeks); the significant reduction of lameness from the pre-surgical status began to become apparent at second follow-up (8 weeks); in addition, the statistical differences between follow-ups show that the evolution of the lameness tends to be progressive over time. This means that part of our second hypothesis, which refers to a faster clinical recovery, is rejected or may need further studies. In this study, the time and results for clinical recovery are similar to previous publications, the restriction on exercise time is also the same as in other studies, and a shorter restriction on exercise time means faster recovery of muscle mass. Thus, a faster clinical improvement as described before, theoretically, if bioresorbable implants had a faster osteointegration, it would be possible to introduce a shorter restriction on postsurgical exercise time.

Complications were observed in 20.8% (5) of the patients that had continued the study for 5 months after surgery (24), attending to the classification proposed by Cook (2010); 2 patients presented minor complications solved with medical treatment, and three presented major complications that needed another surgery [30]. This percentage is within the range of values presented in other papers that studied metallic TTA (11–31.5%) [6,39,64,65,66], and the percentage is lower than the one obtained in another study that used metal-fixed PLA implants [20]. The implant removal was not taken into account in the percentage because it was contemplated on the informed consent as a possibility, although the removal of the pin was carried out in only two dogs (8.3%); in the first patient, the pin was removed for clinical interest, and in the other dog the reason was a lameness without apparent cause, which was solved after the pin removal.

Among the minor complications, one of the dogs presented a fracture of the distal cortical in the tibial crest a month after surgery. This is one of the most common complications in TTA and MMT. There is no displacement; thus, it was solved by strict rest of the patient for a month as recommended in several studies [6,55,65,67,68]. Another minor complication was the appearance of masses, 4 months after surgery, in the area of the incision. They had approximately 1 cm in diameter, the aspect was similar to vesicles with serous liquid inside, they did not present a fistulous path, we carried out radiographs, and no evidence of bone or articular problems was revealed, and the patient had no lameness. We performed a microbiological culture of the liquid in the vesicles, and it was positive for *Staphylococcus pseudintermedius*, so we assumed that the vesicles were a subcutaneous infection following previously published criteria [8], and the treatment was the application of local medication and the administration of antibiotics; the evolution was good, and the vesicles disappeared.

Major complications were observed in three patients, in which cases owners admitted that they did not follow the movement restriction. One of them presented a complicated tibial crest fracture that was solved with surgery by fixing the tibial crest with three pins; nevertheless, the evolution was good, the implant integrated adequately into the bone, the patient solved the lameness and did not need another intervention. Another of the patients had a tension band wiring rupture and underwent surgery for its replacement. Finally, the last patient with a major complication had a cerclage rupture and a folded pin, underwent another surgery to replace it, and the evolution was good.

Regarding the cerclage rupture, several authors reported fracture of the crest or failure of the implant after TTA without significant avulsion of the tibial tuberosity [6,56,57]. That showed the importance of soft tissue in the maintenance of the tibial crest stability [33]. Some authors recommended the replacement of the wire of the conventional titanium plate [33] or placing another cerclage caudal to the first one [69] if necessary.

The rate of infections observed in TTA with metallic implants is 2.6–8.7% [6,8,31,66,70]. Despite the fact that in our study, we did not record any infection, we should mention that the inability of bacteria to persist on a resorbable material is one of the advantages of these over permanent implants [71]. Another author used biodegradable implants on TTA for the replacement of an infected metallic implant maintaining the advancement [21]. Another paper on TTA with metal-fixed biodegradable materials described the case of a patient who suffered an infection; the decision was to eliminate metals [20]. Keeping this in mind, we think that our decision about reducing the use of metals to a minimum was wise despite the loosening of the fixation. In addition, the porous morphology of these implants allows introducing different substances that could fight infections, but further studies are needed in this field.

In human medicine, multiple complications associated with bioresorbable implants were reported, including implant fracture, secondary migration due to poor fixation, aseptic loosening, osteolysis, and chondrolysis [72,73]. We should also point out local acidity that leads to adverse tissue reactions [51]. In our study, we did not record any of these complications, although we may need more studies with a higher number of patients to determine the prevalence of the described complications in dogs.

The limitations of this preliminary study are the number of patients, the lack of a control group, and the lack of objective evaluation methods such as histological examinations, CT scans, or the use of a force platform for lameness assessment. Our aim is to continue investigating and include these exams in future publications.

## 5. Conclusions

The results of this study show that PLA implants for TTA are safe and provide good functional results, comparable to the results obtained with metallic implants.

Secondly, we observed that PLA implants for TTA provide a faster bone healing on the osteotomy gap. In our study, bone healing was not affected by differences in body weight or age.

The results of clinical improvement are similar to those published in other papers.

This technique allows decreasing the economic cost and gives us the possibility to personalize the implants in order to adapt them to every kind of patient.

Moreover, in our experience, MMT is an easier and faster technique than conventional TTA, so we think that the learning curve could be lower.

This is a preliminary study; thus, further studies are needed, with a higher number of patients, as well as achieving a control group.

## Figures and Tables

**Figure 1 animals-11-01271-f001:**
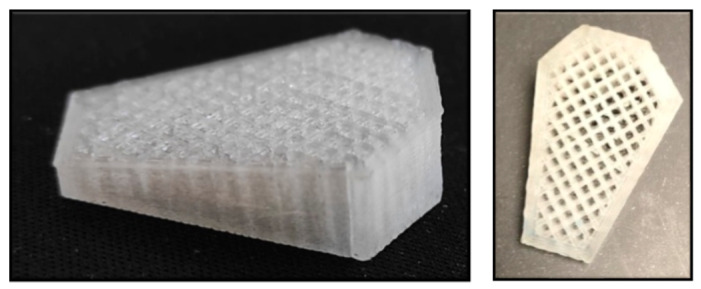
PLA implants.

**Figure 2 animals-11-01271-f002:**
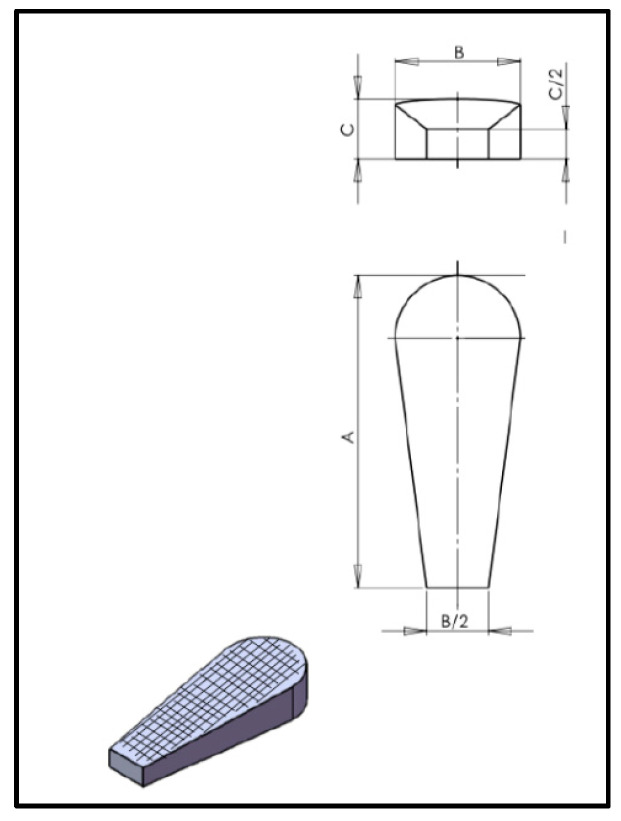
Implant sketch.

**Figure 3 animals-11-01271-f003:**
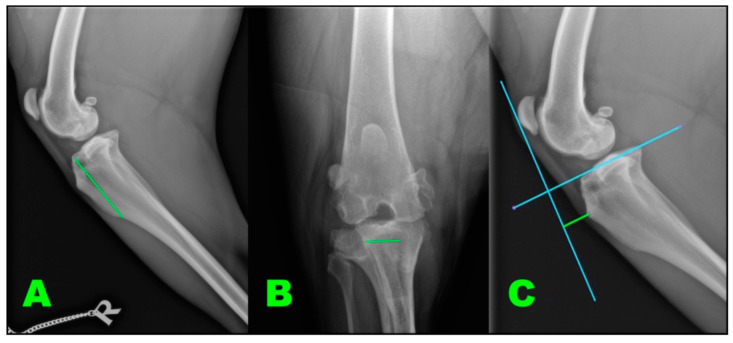
Measures taken to acquire implant conformation. We take three measures (**A**–**C**) corresponding to the green in the images. (**A**) Implant length; (**B**) implant width; (**C**) implant depth.

**Figure 4 animals-11-01271-f004:**
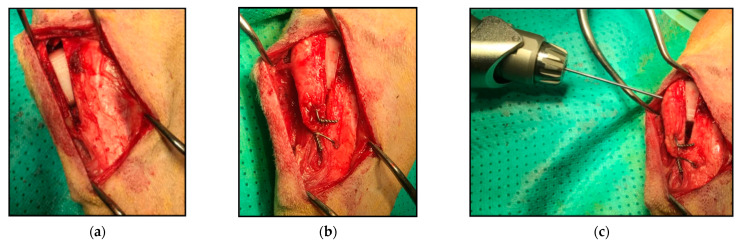
Surgical technique. (**a**) Implant placement. (**b**) Tension band wiring. (**c**) Pin placement. (**d**,**e**) Final result of the technique.

**Figure 5 animals-11-01271-f005:**
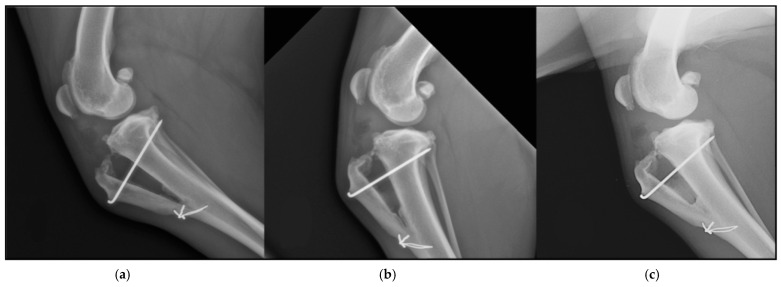
Different times of ossification in a dog. (**a**) Ossification in the first follow-up. (**b**) Ossification in the second follow-up. (**c**) Ossification in the third follow-up.

**Table 1 animals-11-01271-t001:** Mean time for radiologic follow-ups (days).

Mean Time for Radiologic Follow-Ups (Days)		
First follow-up	Second follow-up	Third follow-up
35.22 ± 10.72	71.88 ± 9.39	154.27 ± 22.02

**Table 2 animals-11-01271-t002:** Number of patient per follow-up for ossification degree assessment.

Ossification Degree	First Follow-Up	Second Follow-Up	Third Follow-Up
0	5	0	0
1	6	2	0
2	3	5	0
3	4	9	8
4	0	2	10
Mean	1.33 ± 1.13	2.61 ± 0.84	3.55 ± 0.51

Scale used for the evaluation of the healing of the osteotomy site developed by Hoffmann (2006) [31].

**Table 3 animals-11-01271-t003:** Number of patients distributed by ossification degree in the different follow-ups divided by body weight and age.

Ossification Degree	N of Patient per Follow-Up											
	<30 kg (*n* = 10)			≥30 kg (*n* = 8)			<7 Years (84 Months) (*n* = 12)			≥7 Years (84 Months) (*n* = 6)		
	First	Second	Third	First	Second	Third	First	Second	Third	First	Second	Third
0	4	0	0	1	0	0	4	0	0	1	0	0
1	2	1	0	4	1	0	3	1	0	3	1	0
2	2	4	0	1	1	0	3	5	0	0	0	0
3	2	3	5	2	6	3	2	5	4	2	4	4
4	0	2	5	0	0	5	0	1	8	0	1	2
Mean	1.2 ± 1.22	2.6 ± 0.96	3.5 ± 0.52	1.5 ± 1.06	2.6 ± 0.74	3.6 ± 0.51	1.2 ± 1.13	2.5 ± 0.79	3.6 ± 0.49	1.5 ± 1.22	2.8 ± 0.98	3.3 ± 0.51
Statistically significant differences	No statistically significant differences between groups for the same control time						No statistically significant differences between groups for the same control time					
Mean weight/age	18.72 ± 6.48 kg			36.62 ± 9.62 kg			51.83 ± 21.65 months			112.33 ± 12.27 months		

Scale used for the evaluation of the healing of the osteotomy site developed by Hoffmann (2006) [31]. Consideration for senior dogs ≥ 7 years based on a previous publication by Epstein (2005) [32].

**Table 4 animals-11-01271-t004:** Number of patient per follow-up for lameness degree assessment.

Lameness Degree	Pre-Surgical Assessment	First Follow-Up	Second Follow-Up	Third Follow-Up
0	0	3	9	16
1	0	3	4	2
2	1	5	4	0
3	10	5	1	0
4	3	1	0	0
5	5	1	0	0
Mean	3.5 ± 0.92	2.05 ± 1.39	0.83 ± 0.98	0.11 ± 0.32

Numerical rating scale used for assessing the lameness developed by Etchepareborde (2011) [29].

**Table 5 animals-11-01271-t005:** Complications.

	Complications	
Minor	Fracture of the distal cortical of the tibial crest	1
	Apparition of masses in the incision region	1
Major	Tension band wiring rupture with or without tibial crest displacement	3

Complication attending to the classification proposed by Cook (2010) [26].

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
