# Peer review of "Preliminary Clinical and Radiographic Evaluation of a Novel Resorbable Implant of Polylactic Acid (PLA) for Tibial Tuberosity Advancement (TTA) by Modified Maquet Technique (MMT)"

_animals, 2021, doi:10.3390/ani11051271_

Round 1

Reviewer 1 Report

In order to better understand the possible complications of the described technique, I recommend a more detailed explanation of the technique, with special attention to the pin implantation technique.
It would be interesting to explain how to avoid rotational movements of the pin.
It would also be interesting to describe the technique of placing the tension band. 

Addition:

  1. What is the main question addressed by the research?

The author proposes the application of new materials for the resolution of cranial cruciate ligament rupture, in the TTA technique (tibial tuberosity advancement), using the modified Maquet technique (MMT), mainly comparing the ease of its implementation with the results obtained and complications, compared to techniques that use metal implants for its resolution.

  1. Do you consider the topic original or relevant in the field, and if so, why?

I considered it important, and of great relevance, as it is about providing new alternatives with the use of biocompatible materials for the resolution of one of the most frequent pathologies in canine orthopaedics, such as cranial cruciate ligament rupture (RLCA).

More and more research is being conducted into the use of biocompatible and biodegradable materials in orthopaedic surgery, with very good results and sometimes less traumatic than the use of traditional implants.

  1. What does it add to the subject area compared with other published material?

Many techniques have been described to date for the resolution of this pathology.

In my opinion, apart from the characteristics of the material, already presented in other studies, it seems relevant to me the personalisation in each case, through the application of 3D technology, based on the measurements obtained radiologically in each patient.

  1. What specific improvements could the authors consider regarding the methodology?

I think that the methodology of the study is correct, but as they say in their study a larger number of patients is needed to draw clearer conclusions, but as a preliminary study it is very adequate.

Regarding the materials and methods, the description of the technique should perhaps be a little more descriptive.

  1. Are the conclusions consistent with the evidence and arguments presented and do they address the main question posed?

Without a doubt, in my opinion, this is a very well-designed preliminary study for the results they wanted to analyse.

  1. Are the references appropriate?

It is a well-revered study, with an abundant bibliography that gives a good overview of the state of research in this field in small animals.

  1. Please include any additional comments on the tables and figures.

Probably larger images with posterior anterior projection would give us a clearer view of the technique used and the final result.

Regarding the tables, they seem to me to be correct for this study.

Author Response

Dear reviewer, 

Thank you very much for the suggestions and reviews. We consider that they will be very useful to increase the quality of the publication and its understanding. In the manuscript, the information that has been added and modified is underlined in yellow. I attached a PDF with a more extensal answer to your suggestion, please see the attachment

Kind regards

Reviewer 2 Report

The paper describes a randomized clinical trial that aims to evaluate the effects of using PLA biomaterial as cage for the TTA surgery. The results were evaluated in terms of surgical success, lameness improvement and X-rays bone healing rate.

The work is formally correct, the data analysis is appropiate and the results are well described, but there are some weakness.

  1. The first point regards the design of the study, infact, in a clinical randomized trial it's necessary to perform the power analysis and consequentely calculate the sample size of the study.  This aspect isn't mentioned in your paper, did you do it?
  2.  In the introduction, in lines 61-64, you state that "In addition to that, metallic implants cause higher stiffness than bone tissue, as well as mechanical stress shielding in bone inducing its resorption, ultimately leading to osteolysis or prosthesis loosening" this isn't totally correct, indeed there are many metallic alloy, such as Porous titanium alloy that mimicking bone trabecular structure and reduce the stress shielding. I suggest you include this aspect in the introduction to make it more complete. I am attaching a bibliographic reference but there are many others in the scientific literature. (Zhang C, Zhang L, Liu L, Lv L, Gao L, Liu N, Wang X, Ye J. Mechanical behavior of a titanium alloy scaffold mimicking trabecular structure. J Orthop Surg Res. 2020 Feb 7;15(1):40. doi: 10.1186/s13018-019-1489-y. PMID: 32028970; PMCID: PMC7006186.)
  3. What method did you use to measure tibial advancement? from figure 3c it seems to me that you have used the conventional method, called Tibial Plateu Angle (TP-PTA), but it isn't mentionated. Additionally, from figure 3c, it seems that there are some errors in the mesurement, in fact the line passing throught the tibial plateau should be perpendicular to the line starting from the cranial margin of the patella. I'm attaching a reference related to this aspect. (Millet M, Bismuth C, Labrunie A, Marin B, Filleur A, Pillard P, Sonet J, Cachon T, Etchepareborde S. Measurement of the patellar tendon-tibial plateau angle and tuberosity advancement in dogs with cranial cruciate ligament rupture. Reliability of the common tangent and tibial plateau methods of measurement. Vet Comp Orthop Traumatol. 2013;26(6):469-78. doi: 10.3415/VCOT-13-01-0018. Epub 2013 Oct 22. PMID: 24146134.
  4. an important weakness of the work is represented by the lack of objective evaluations, especially as regards the evaluation of the progress of the lameness. You have in fact used an evaluation scale which is still a subjective evaluation operator-dipendent.
  5. At line 373 you state that "100%  of the dogs had no detectable lameness at the last follow-up, 88.89% with normal weight bearing at stance". How did you evaluate the weight bearing? Considering points 4 and 5, I believe that it would have been better to use objective mesurement system for gait and/or stance analysis. This weakness reduces the scientific validity of the study. 
  6. As you said, the lack of a control group is another weakness of your study.

Minor consideration:

  •  Line 320: adjust bibliografic note.

Considering this, I believe that this paper could be suitable for the pubblication after making the suggested corrections.

Author Response

Dear reviewer, 

Thank you very much for the suggestions and reviews. We consider that they will be very useful to increase the quality of the publication and its understanding. In the manuscript, the information that has been added and modified is underlined in yellow. I attached a PDF with a more extensal answer, point by point, to your suggestion, please see the attachment

Kind regards
